# Potential Effect of Baobab’s Polyphenols as Antihyperlipidemic Agents: In Silico Study

**DOI:** 10.3390/molecules28166112

**Published:** 2023-08-17

**Authors:** Alaa Alnoor Alameen, Monerah R. Alothman, Mona S. Al Wahibi, Ejlal Mohamed Abdullah, Rehab Ali, Mohnad Abdalla, Sndos Z. A. Fattiny, Rasha Elsayim

**Affiliations:** 1Department of Pharmacology and Toxicology, College of Pharmacy, King Saud University, P.O. Box 2457, Riyadh 11451, Saudi Arabia; aalameen1.c@ksu.edu.sa; 2Department of Botany and Microbiology, College of Science, King Saud University, P.O. Box 2455, Riyadh 11451, Saudi Arabiamalwahibi@ksu.edu.sa (M.S.A.W.); 3Department of Biochemistry, College of Science, King Saud University, P.O. Box 2455, Riyadh 11451, Saudi Arabia; eishag.c@ksu.edu.sa; 4Department of Drug and Toxicology, College of Pharmacy, King Saud University, P.O. Box 2457, Riyadh 11451, Saudi Arabia; reali@ksu.edu.sa; 5Department of Medicine, Vascular Biology Center, Medical College of Georgia at Augusta University, Augusta, GA 30912, USA; mohnadabdalla200@gmail.com; 6Department of Food Science and Nutrition, College of Food and Agriculture Sciences, King Saud University, P.O. Box 2460, Riyadh 11451, Saudi Arabia; 442204496@student.ksu.edu.sa

**Keywords:** hyperlipidemia, baobab, polyphenols, molecular docking, dynamic simulation

## Abstract

*Adansonia digitata* L. is an African tree commonly called baobab. This tree is effectively used in traditional medicine to treat cardiovascular disorders. Hyperlipidemia is a well-known cardiovascular risk factor associated with the increased incidence of mortality worldwide. This study aimed to demonstrate the mechanism of baobab polyphenols in the activities of hydroxy-3-methylglutaryl coenzyme A (HMG-CoA) reductase and pancreatic lipase as lipid metabolic enzymes. Molecular docking and an incentive for drug design showed that all the polyphenols in baobab bound to the proteins with higher affinity and a lower binding energy compared with simvastatin as the positive control (ΔG: from −5.5 kcal/mol to −6.5 kcal/mol). The same polyphenols exhibited a considerable binding affinity to pancreatic lipase (ΔG: from −7.5 kcal/mol to −9.8 kcal/mol) in comparison with the control and HMG-CoA reductase. Quercetin showed the best docking score from the selected Baobab polyphenols (ΔG = −9.8 kcal/mol). The root mean square deviation (RMSD) results indicated that stable epicatechin and quercetin complexes were demonstrated with HMG-CoA reductase, and other less stable complexes were developed using rutin and chlorogenic acid. Moreover, the analysis of the root mean square fluctuation (RMSF) simulation results was consistent with that of the RMSD. The RMSF value for all the baobab polyphenols, including the crystal control ligand, was kept between 0.80 and 8.00 Å, similarly to simvastatin, and less than 4.8 Å for pancreatic lipase. Chlorogenic acid, quercetin, epicatechin, and rutin had negative ΔG binding scores from highest to lowest. The same ligands displayed more negative ΔG binding scores than those observed in HMG-CoA reductase and crystal control ligand (methoxyundecyl phosphinic acid) in their simulation with pancreatic lipase. In conclusion, baobab polyphenols interact with HMG-CoA reductase and pancreatic lipase to inhibit their substrate binding and block their activity.

## 1. Introduction

Hyperlipidemia, especially that which includes elevated low-density lipoprotein and cholesterol levels, is a well-known cardiovascular risk factor associated with the increased incidence of coronary heart disease, which is considered one of the major causes of mortality worldwide [1,2].

Several enzymes, including pancreatic lipase and 3-hydroxy-3-methylglutaryl coenzyme A (HMG-CoA) reductase, are involved in lipid metabolism. HMG-CoA reductase is essential in cholesterol biosynthesis; it converts the HMG-CoA substrate to mevalonate, which leads to cholesterol synthesis via a series of biochemical reactions. Therapeutically, the inhibition of this enzyme by drugs, such as statins, lowers cholesterol synthesis and therefore decreases circulating cholesterol levels [3]. Statins inhibit HMG-CoA reductase by binding to its active site and prevent the conversion of its substrate to cholesterol [4]. Lopez LM documented that 61% of CAD patients not treated with antihyperlipidemic drugs did not meet the target cholesterol levels set by the National Cholesterol Education Program [5]. Pancreatic lipase assists in the synthesis of fatty acids by hydrolyzing triglycerides. Food and Drug Administration-approved drugs, such as orlistat, reduce fat absorption in humans by inhibiting pancreatic lipase [6]. Orlistat is not well tolerated due to its unfavorable side effect profile and high discontinuation rate among users; statins have been linked to the development of type 2 diabetes mellitus and autoimmune diseases when used long term [7]. Therefore, alternative agents with low toxicity and high tolerance are required for future therapeutics of dyslipidemia. Nutritional modifications associated with medical therapy have attracted attention as novel treatments to alleviate dyslipidemia. In recent years, dietary polyphenols have received much attention regarding the prevention of diseases such as type 2 diabetes, osteoarthritis, obesity, hyperlipidemia, and hyperuricemia [8]. More specific animal studies showed that polyphenols could decrease the plasma level of low-density lipoprotein-cholesterol (LDL-C l) [9]. Moreover, in the past three decades, numerous researchers have shown interest in dietary fats and their involvement in cardiovascular diseases (CVDs). The low risk of CVD in those who consume high amounts of polyphenol, regardless of a high-cholesterol and high-fat diet, is a sign of the beneficial effects dietary polyphenols have in the maintenance of cardiovascular health [10].

Baobab (*Adansonia digitate* L.) is considered an important multipurpose food tree in African countries and has been well known as a traditional medicine for centuries. Different parts of this plant possess interesting pharmacological and medicinal activities, including antioxidant, [11] antimicrobial, [12] anti-inflammatory and analgesic, [13] antipyretic, [14] and antidiarrheal activities [14]. Phytochemical analysis of baobab revealed that it contains phenolic acids, flavonoids, tannins, saponins, triterpenes, citric acid, and other constituents [11].

Although several previous in vivo and in vitro and current clinical trials have revealed that the administration of baobab extract lowers cholesterol levels and improves dyslipidemia, limited investigations have focused on its bioactive compound(s) and their precise pharmacological and molecular mechanisms as antihyperlipidemic agents. Since polyphenols (in form of phenolic acid and flavonoids) are the most abundant constituents among others, we focused on them and proposed that the lipid-lowering activity of baobab might be due to its polyphenolic content. Therefore, this study aimed to demonstrate the mechanism of baobab polyphenols on the activity of HMG-CoA reductase and pancreatic lipase by molecular docking and provides an incentive for drug design. We investigated the known polyphenols present in baobab from previous studies and conducted running docking studies to screen and select those that exhibited the highest binding score and energy for further molecular dynamic (MD) simulation. We used simvastatin and methoxyundecyl phosphinic acid (MUP) as control inhibitors of HMG-CoA reductase and pancreatic lipase, respectively.

## 2. Results and Discussion

### 2.1. Molecular Docking Study

The cocrystal control (simvastatin) bound the HMG-CoA reductase with high affinity, and the three amino acid residues (GLU 559, LYS735, and ASN 755) formed H- and ionic bonds. All the polyphenols in baobab bonded to this protein with high affinity, which is known from their lower binding energy comparable to the simvastatin positive control (ΔG range from −5.5 kcal/mol to −6.5 kcal/mol) in Table 1. The first polyphenol that bonded to the HMG-CoA reductase with considerable affinity was quercetin (ΔG = −6.2 kcal/mol). It formed H-bonds with GLY 560, HIS 752, and ALA 564 in the target protein. Similarly, rutin displayed an even stronger binding affinity to the HMG-CoA reductase than quercetin (ΔG = −6.5 kcal/mol for both) with an H-bond formed with GLU 559, ALA 564 and ALA 751 residues of HMG-CoA reductase. This result was similar to the computational analysis conducted by Arbianti et.al., who found that quercetin displayed the best affinity score for the inhibition of the HMG-CoA reductase enzyme (−7.8 kcal/mol) against simvastatin and the other compounds that were studied [15].

On the other hand, the same polyphenols showed considerable binding affinity to pancreatic lipase (ΔG ranged from −7.5 kcal/mol to −9.8 kcal/mol) in comparison with the control and HMG-CoA reductase (Table 1). This result is in agreement with that of Cicolari et al., who reported that the baobab hydromethonolic extract exhibited potent inhibitory activity against pancreatic lipase in comparison with HMG-CoA reductase [16]. Quercetin showed the best docking score from the selected baobab polyphenols (ΔG = −9.8 kcal/mol) and bound to the ASP 79, ARG 256, GLY76, and HIS 151 residues in pancreatic lipase. The remaining polyphenols, namely, epicatechin, rutin, and chlorogenic acid, bound to pancreatic lipase with considerably better affinity than with the cocrystal control of the protein.

Figure 1A,B present the interactions between the baobab polyphenols and the target proteins of the HMG-COA reductase and pancreatic lipase in 2D and 3D structures, respectively. They showed differences in the binding of the baobab polyphenols under study to both enzymes entrenched in the coordination of the polyphenol rings and the side chains present on them. To block HMG-CoA reductase activity (from reducing the substrate HMG-CoA to cholesterol), the inhibitor must compete with HMG-CoA to reach its appropriate positioning and bind to SER684, ASP690, LYS691, and LYS692 (essential residues to bind the HMG moiety) and GLU559 and HIS866 (important residues in binding the CoA moiety) [8]. Similar to simvastatin, all the polyphenols, except quercetin, bound to one of these essential residues through strong H-bonding but did not exhibit full occupancy. This is in agreement with the report that indicated that polyphenol rings form rigid bonds and thus cannot completely fill the binding site of the protein and hence might not act as competitive inhibitors of HMG-CoA reductase [10].

On the other hand, the SER152, ASP176, AND HIS263 amino acid residues in the active pocket of pancreatic lipase maintained their hydrolytic activity. Figure 1 shows that the MUP crystal control inhibitor of the enzyme bound to the PHE77 and HIS 263 residues of the enzyme, and the baobab polyphenols behaved similarly by binding to either one or both of these residues.

### 2.2. MD Simulation Study

Given that docking treats proteins as fixed molecules and ignores the time spent processing ligand–protein interactions, we confirmed the docking results, which indicated the interaction between the baobab polyphenols and the target proteins under study, by performing MD simulation. This method has advantages such as the capability to measure the rigidity of protein–ligand complexes under specific simulated body situations and to establish the stability of target proteins in their combination with the desired polyphenols.

### 2.3. Stability of Protein–Ligand Complexes

Two qualitative indicators—the root mean square deviation (RMSD) and root mean square fluctuation (RMSF)—were employed to describe the stability of the protein–ligand complexes. The values of RMSD showed that the ligands that were most stable in their interactions with the protein, and those of RMSF revealed the compactness of the complex measured by their fluctuations [17,18]. The presence of widespread fluctuation in the trajectories of RMSD and RMSF indicates configurational changes in the protein–ligand interaction [19]. The ligand–protein complexes were analyzed for 100 ns to measure RMSD and RMSF.

With respect to the HMG-CoA reductase complexes with the baobab polyphenols, the present study’s results indicated a stable system for all the polyphenols except chlorogenic acid (Figure 2). The best RMSD behavior was observed for the epicatechin–HMG-CoA reductase complex (Figure 2A), with slight fluctuations (3.00–12.00 Å) observed at two points at the beginning of the trajectory and at approximately 30 ns for the ligand and then stabilized at equilibrium until the trajectory ended. Meanwhile, the HMG-CoA reductase protein showed fluctuations between 2.00 and 8.00 Å. By contrast, chlorogenic acid displayed a major fluctuation (2.00–64.00 Å), and its protein showed narrow fluctuations between 1.6 and 6.4 Å (Figure 2B).

Quercetin exhibited a behavior similar to that of the control ligand (simvastatin), with narrow fluctuations at several time points, namely, 10, 40, and 60 ns (2.00–15.00 Å), and then stabilized within a fluctuation range of 4.00–12.00 Å (Figure 2C). The protein fluctuated at the initial phase of the simulation (2.00–6.00 Å) and then equilibrated within the fluctuation range of 4.00–7.00 Å until the end of the simulation. The RMSD of rutin in the ligand–protein complex (Figure 2D) demonstrated slight fluctuations (3.00–16.00 Å) up to 20 ns and then exhibited a stable state of fluctuation (10.00–14.00 Å) until the end of the simulation. The protein in the rutin–HMG-CoA reductase system during the simulation period demonstrated consistent fluctuation between 3.00 and 9.00 Å.

Overall, epicatechin and quercetin displayed steadily stable complexes with HMG-CoA reductase, whereas less stable complexes were revealed by rutin and cholorogenic acid. This result looks similar to that obtained by Riyad et.al., who concluded that quercetin formed a stable complex with HMG-CoA reductase, with RMSD values fluctuating between 0.5 and 2.2 Å [20]. On the other hand, the baobab polyphenols maintained a different behavior from the cocrystal control (MUP) when simulated with pancreatic lipase (Figure 3). Epicatechin, cholorogenic acid, and rutin (Figure 3A,B,D) showed narrow fluctuation ranges of 0.8–3.00, 5.0–8.5, and 2.4–5.4 Å, respectively. However, both quercetin and MUP (Figure 3C,E, respectively) possessed massive fluctuation ranges (2.0–16.0 Å for quercetin and 5.0–40.0 Å for the crystal control ligand). Pancreatic lipase fluctuated from 1.0 Å to approximately 4.0 Å in its simulation with epicatechin, chlorogenic acid, rutin, and MUP, but showed a higher RMSD with quercetin (up to 7.5 Å) (Figure 3).

During the simulation period, the RMSF was used to measure the extent of the movement or fluctuation of each residue [21]. As shown in Figure 4, the analysis results of the RMSF and RMSD simulation were consistent and confirmed each other. The RMSFs of all the baobab polyphenols were maintained between 0.80 and 8.00 Å in the same range as those of simvastatin, which was the cocrystal control (Figure 4A–E). The RMSF was less than 4.8 Å for all the complexes between the pancreatic lipase and polyphenols, including the crystal control ligand (Figure 5A–E).

The broad oscillations in RMSD and RMSF were possibly caused by perturbations in the system, the repositioning of ligands within the binding site, and conformational changes within the protein complex system [19,22]. This outcome showed strong associations with the binding energies of the complexes (Table 1).

To calculate the free energy of polyphenols bound to HMG-CoA reductase, we conducted molecular mechanics with generalized Born and surface area solvation (MM/GBSA). In the MM-GBSA analysis, the best score of the ΔG binding was represented by the most negative ΔG binding score (the lowest) [18,23]. Compared with the MM-GBSA binding energies of the control ligand (−31.62 kcal/mol), the complexes of epicatechin and rutin exhibited more negative ΔG binding scores in the following order: rutin < epicatechin < quercetin < chlorogenic acid. The outcomes showed that chlorogenic acid had the lowest binding affinity to HMG-CoA reductase and the maximum binding energy (Table 2A). In the comparison of the MM-GBSA values of HMG-CoA reductase (Table 2A) with those of pancreatic lipase (Table 2B), the same ligands had the best ΔG binding score, that is, they displayed a more negative ΔG binding score (down to −57.9 kcal/mol) than HMG-CoA reductase and the crystal control ligand (MUP). We can arrange the ligand–protein complexes based on their ΔG binding scores as follows: quercetin < rutin < chlorogenic acid < epicatechin. MUP exhibited minimum binding affinity to pancreatic lipase and maximal binding energy (Table 2B).

### 2.4. Protein and Ligand Properties from MD Simulation Analysis

We evaluated the values of different parameters, namely, the radius of gyration (rGyr), polar surface area (PSA), solvent-accessible surface area, ligand RMSD, and molecular surface area (MolSA), to assess the ligand features that indicate ligand behavior inside the binding pocket of the target enzymes. The structural stability of the ligand–enzyme system and anticipated conformational changes were both revealed by the RMSD. The RMSD values resulting from the simulation analysis of the baobab polyphenols and HMG-CoA reductase showed that epicatechin and quercetin (Figure 6A,C, respectively) remained at equilibrium between 0.3 and 1.5 Å, which are below the values displayed by the cocrystal ligand inside the protein (0.5 Å to 2.4 Å), during the entire simulation. Chlorogenic acid and rutin (Figure 6B,D) had higher RMSD values of approximately 0.5 Å to 2.4 Å and 1.5 Å to 4.5 Å, respectively. Given that a low RMSD value (lower than 2 Å) indicates an improvement in the binding pose, epicatechin and quercetin displayed the best structural stability among the polyphenols.

On the other hand, simulation analysis of the same polyphenols inside pancreatic lipase showed a RMSD of 1.00–2.00 Å for chlorogenic acid (Ligand B) and rutin (Ligand D) and 0.3–1.5 Å with equilibrium at 0.5 Å for epicatechin (Ligand A) (Figure 7). The three polyphenols mentioned previously showed analogous simulations displayed by the cocrystal control ligand (MUP), which fluctuated between 0.8 and 2.4 Å. However, the last ligand (C), quercetin, showed different behavior and massive fluctuations ranging from 25 Å to 60 Å with equilibrium at 50 Å (Figure 7).

Another ligand property analyzed was the rGyr, which indicates the protein compactness conformation and protein folding characteristics [18]. A low value of rGyr indicates high compactness, but a high value points to a low attachment of the compound to the protein. The results demonstrated that the baobab polyphenols exhibited a similar behavior to simvastatin in their simulation with HMG-CoA reductase, with small variations within a reasonable range gradually reaching a state of balance. The candidate ligands had rGyr values of 3.68–3.84 (equilibrium at 3.76 Å), 4.0–4.8 (equilibrium at 4.4 Å), 3.7–3.80 (equilibrium at 3.75 Å), and 4.8–5.5 Å (equilibrium at 5.0 Å) for epicatechin, chlorogenic acid, quercetin, and rutin, respectively, in their simulation with pancreatic lipase (Ligands A–D in Figure 6). In comparison, the simulation of the same ligands with pancreatic lipase revealed values similar to those observed with the cocrystal control ligand (MUP) and the HMG-CoA reductase simulation trajectory. The rGyr values were 0.6–0.8 (equilibrium at 0.75 Å), 4.0–4.8 (equilibrium at 4.8 Å), and 3.66–3.80 (equilibrium at 3.74 Å), and 4.2–4.8 Å (equilibrium at 4.4 Å) for epicatechin, chlorogenic acid, quercetin, and rutin, respectively (Ligands A–D in Figure 7). The simulation trajectory’s stable rGyr values demonstrated that HMG-CoA reductase and pancreatic lipase were firmly folded and kept compressed.

The MolSA of the ligands fluctuated most of the time at varying ranges. Rutin and simvastatin (Ligands D and E in Figure 6) presented the highest MolSA in their complexes with HMG-CoA reductase (between 480 and 420 Å^2^, respectively). Meanwhile, epicatechin, chlorogenic acid, and quercetin (Ligands A–C in Figure 6) demonstrated reduced MolSA and fluctuations (26, 312, and 258 Å^2^, respectively). Notably, rutin (Figure 7D) displayed high MolSA values when simulated with pancreatic lipase (485 Å^2^), and the other polyphenols exhibited similar values (Figure 7).

The SASA values for all the ligands fluctuated within 150–500 Å^2^ in their simulation with both proteins. An exception was quercetin, which possessed a lower SASA value when simulated with pancreatic lipase (25–80 Å^2^) (Ligand C in Figure 6 and Figure 7). On the other hand, depending on the computational analysis, we can arrange the baobab polyphenols based on their PSA values in their simulation with the proteins of HMG-CoA reductase and pancreatic lipase as follows: rutin > chlorogenic acid > quercetin > epicatechin. All the ligands exhibited higher PSA values than the cocrystal control ligand (Figure 6 and Figure 7). Given that these numbers give an estimation of the blood–brain barrier (BBB) permeability, an applicant who meets the PSA limit of 90Å^2^ or less may be targeted for the central nervous system [24], which means that all the baobab polyphenols investigated in the present in silico study cannot cross the BBB.

### 2.5. Protein–Ligand Interaction and Bond Formation

Two covalent or noncovalent bonds were observed during the ligand–protein interactions. The most widely recognized bonds were those involving hydrogen (H-bonds), hydrophobic interactions, ionic strength, van der Waals forces, and solvent bonds (water bridges). Given that H-bonds are the major component in ligand binding, they were considered during computational analysis and the drug design process. Several amino acid residues of both target proteins interacted with the candidate polyphenols in hydrophobic interactions, H-bonding, and other interaction types. The essential amino acid residues in the reaction of HMG-CoA reductase with its substrate (baobab polyphenols) through H-bond formation were ALA556, GLU559, SER565, HIS752, and ASN755 with epicatechin with interaction fractions of 18%, 80%, 25%, 20%, and 18%, respectively, during the trajectory (Figure 8A). The residues for the interaction with chlorogenic acid included CYS561, ALA564, SER565, ASN567, ARG567, ARG571, LYS722, and SER852, with 10%, 10%, 15%, 30%, 40%, 20%, and 18% interaction fractions, respectively (Figure 8B). Meanwhile, GLU559, GLY560, SER565, ARG568, ALA751, HIS752, ASN755, and SER852 interacted with quercetin, with interaction fractions of 25%, 13%, 15%, 10%, 10%, 20%, and 12%, respectively (Figure 8C). GLU730, ASN734, LEU857, and ALA858 accounted for 95%, 20%, 18%, and 18% interaction fractions with rutin, respectively (Figure 8D). This result is in agreement with the computational analysis of rutin with HMG-CoA reductase that showed its binding with key residues of the protein with stable and strong H-bonds throughout the simulation trajectory [25]. However, only three amino acids (LYS735, HIS752, and ASN755) bounded with H-bond to simvastatin with 12%, 5%, and 30% fractions (Figure 8E).

Concerning hydrophobic bonds, quercetin and simvastatin bound to the residues of HMG-CoA reductase, namely, CYS561, LEU562, HIS752, LEU853, ALA856, and LEU857, with 12%, 10%, 40%, 33%, 8%, and 4% interaction fractions for quercetin, respectively (Figure 8C), and 5%, 1%, 20%, 32%, and 3% for simvastatin (Figure 8E). Other residues formed van der Waals bonds with the following remaining ligands: CYS561 and HIS752 with epicatechin; CYS561, VAL563, ALA564, ARG568, and LEU853 with chlorogenic acid; and VAL731 and LEU857 with rutin (Figure 8B,D). In addition, although the control ligand exhibited multiple ionic bonds with a number of residues, limited ionic interactions with low interaction fractions were displayed by the baobab polyphenols, including ARG571 and LYS722 with chlorogenic acid (Figure 8B), GLU559 and ARG568 with quercetin (Figure 8C), and GLU730 and LYS735 with rutin (Figure 8D). Moreover, ionic bonds were missing in epicatechin (Figure 8A). The water bridge was the last type of bond, where protein surface residues promoted bond organization and enhanced the transfer of electrons between our target proteins and ligands. Chlorogenic acid and quercetin interacted with the greatest number of HMG-CoA reductase residues, which also had H-bonding as mentioned previously. Meanwhile, the same residues that exhibited H-bonds with the studied ligands also facilitated binding to epicatechin, rutin, and simvastatin through the water bridge (Figure 8A–E).

By contrast, when MD simulations were repeated with pancreatic lipase, a limited number of residues interacted with our candidate ligands. However, the control ligand, MUP, bound to numerous residues. MUP formed hydrogen bonds at different ratios, with interaction fractions of less than 3% with the following residues: TYR5, HIS26, PHE77, THR112, SER152, and ALA259 (Figure 9E). Meanwhile, TYR5, PRO24, VAL34, PHE77, ILE78, TYR114, PRO180, ILE209, LEU 213, PHE215, TRP252, PHE258, ALA259, ALA260, and LEU264 were involved in hydrophobic bonding with MUP, with interaction fractions of less than 12%. The water bridge was observed with THR21, GLU22, PHE77, ASP79, ARG111, THR112, SER152, ALA259, and HIS263, with interaction fractions of up to 17.5%. Computational analysis revealed that GLY76, PHE77, ASP79, TYR114, SER152, PHE 215, and HIS263 residues linked to ligands through H-bonding at different interaction fractions, with epicatechin displaying the highest contact percentage of 98% with ASP79 in the trajectory (Figure 9A). PHE77, ILE78, TYR114, PHE215, ALA260, HIS263, and LEU264 exhibited hydrophobic interactions, with quercetin displaying the highest interaction fractions with most of the residues (Figure 9C). The following pancreatic lipase residues facilitated water bridge binding with the baobab polyphenols: GLY76, PHE77, ILE78, ASP79, HIS151, SER152, ASP176, ASP205, TYR255, ARG256, PHE258, and ALA259. With respect to ionic interactions, this bond was observed only with chlorogenic acid, which interacted with two residues in small fractions (less than 2% (Figure 9B)).

Although the results of the computational analysis established that all the baobab polyphenols showed attractive interactions with diverse amino acids in either HMG-CoA reductase or pancreatic lipase in different interaction fractions, the strength of these bonds must be assessed by observing the interaction between the proteins and ligands within the simulation trajectory. Regarding the quantity of HMG-CoA reductase contacts, the results showed that up to six contacts were observed with simvastatin and rutin (Figure 10D,E, respectively). Meanwhile, epicatechin, chlorogenic acid, and quercetin displayed up to nine contacts throughout the simulation (Ligands A–C in Figure 10, respectively). Deep continuous bands with ALA556, THR557, GLU559, CYC561, and LEU562 were observed for epicatechin alone (Figure 10A). By contrast, deep disturbed bands were detected for the remaining polyphenols and simvastatin (Figure 10B–E). As shown in Figure 10, some of the aforementioned amino acid residues interacted with these ligands more favorably than the others, and we can arrange the candidate polyphenols based on their stability of interaction with HMG-CoA reductase as follows: epicatechin > chlorogenic acid > rutin > quercetin. However, the control ligand, simvastatin, displayed a weak ligand–protein contact.

On the other hand, although the simulations were stable throughout the majority of the trajectories (100 ns) for our candidate ligands in their complexes with pancreatic lipase, the control ligand exhibited the weakest contacts. Deep continuous or interrupted bands were detected for PHE77, ASN79, TYR114, SER152, PHE215, HIS263, LEU264, and ALA259 with the following polyphenols: quercetin > rutin > epicatechin > chlorogenic acid (Figure 11A–D).

Overall, the interaction between the baobab polyphenols and candidate proteins prevented the completion of the active site of both enzymes, which inhibited their substrate binding and consequently blocked the activity of the HMG-CoA reductase and pancreatic lipase.

## 3. Methodology

### 3.1. Protein and Ligand Preparation and Docking

The two-dimensional (2D) structures of epicatechin, chlorogenic acid, quercetin, rutin, simvastatin acid, and MUP were downloaded from the PubChem online database (https://pubchem.ncbi.nlm.nih.gov, accessed on 11 December 2022) (PubChem IDs: 72276, 1794427, 5280805, 5280343, 54454, and 446977). With the use of PyMOL and MOE, images of HMG-CoA reductase and pancreatic lipase–colipase were created. Images of these compounds underwent geometric optimization and were stored in [.sdf] format. The 2D structural compounds were then converted to 3D structures and stored in [.sdf] format. The Protein Data Bank database (https://www.rcsb.org/ accessed on 11 December 2022) was used to download the HMG-CoA reductase and pancreatic lipase–colipase proteins, with IDs 1HW9 and 1LPB, respectively. High binding conformations and complex orientations were docked with HMG-CoA, and Auto Dock Vina (version 4; The Scripps Research Institute, La Jolla, CA, USA) was used to employ pancreatic lipase–colipase. The visualization of amino acid residues was carried out using BIOVIA Discovery Studio Visualizer 2020.

### 3.2. MD Simulation

The MD simulation used Desmond with standard methods [26]. The protein was solvated in Desmond System Builder using the TIP3P model. NaCl (0.15 M) was used to neutralize the simulation system. At a pressure of 1.013 bar and a temperature of 310 K, the simulation was run for 100 ns. The trajectory was examined using Desmond, VMD, and PyMOL [16].

## 4. Conclusions

People who take medications for hyperlipidemia usually experience side effects, which often affect the duration of the medication. Thus, alternative options are needed. Plant extracts are a source of a wide range of physiologically active substances with limited side effects, and traditional medicine serves as a foundation for the development of novelties in drug discovery. Baobab extract has been popular among local African people for its efficacy in the depletion of hyperlipidemia, and thus it offers a high probability of including various components that may target different lipid metabolic pathways. We investigated the probable biological effect of the polyphenolic compounds found in baobab on the actions of HMG-CoA reductase and pancreatic lipase.

In this study, quercetin demonstrated the best docking scores with HMG-CoA reductase and pancreatic lipase. Phenolic compounds may inhibit enzyme activity either by forming H-bonds with enzyme residues, which may modify the structure of the enzyme and reduce its activity, or by occupying the active site and blocking it. Rutin has a considerable affinity for the enzymes’ active site. However, the rigid phenolic ring prevents full site occupancy, but it may still block and prevent the substrate from binding. All the polyphenols in this study presented high affinities to pancreatic lipase and showed better results than the control or HMG-CoA reductase. Then, we conducted MD simulations to confirm and measure the stability and rigidity of the protein–ligand complexes. Epicatechin and quercetin formed the most stable complexes with HMG-CoA reductase. The simulated pancreatic lipase exhibited narrow fluctuations during the binding with epicatechin, chlorogenic acid, and rutin. An MM-GBSA study was performed, and epicatechin and rutin exhibited more negative ΔG binding scores than the other phenols. This study showed that baobab has promising potential as a source of active phenolic compounds that can be used as pharmaceutical products. Further in vitro and in vivo research is needed to emphasize how these phenolic chemicals affect the activity of enzymes. In addition, by utilizing a unique inverse molecular docking technique, we found other possible lipid metabolic protein targets for *Adansonia digesta* L. polyphenols [25]. The results may further illuminate their potential as anticancer or antihyperlipidemic agents.

## Figures and Tables

**Figure 1 molecules-28-06112-f001:**
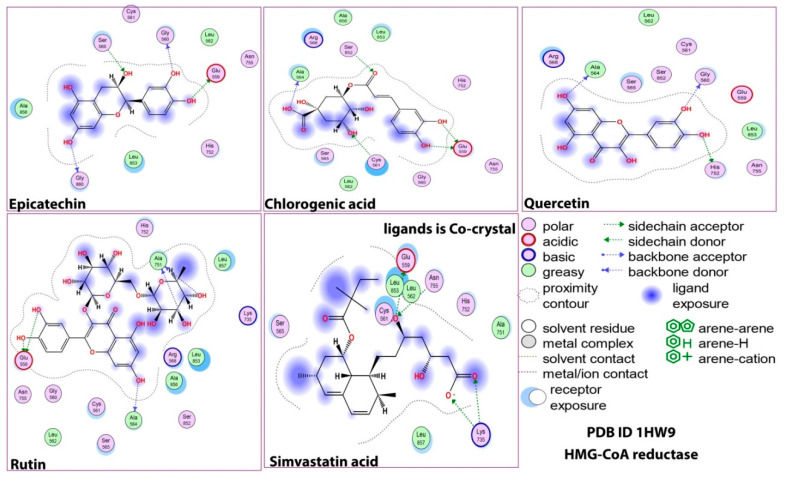
(**A**) Interactions between baobab polyphenols and the target proteins HMG-COA reductase and pancreatic lipase in 2D structures. (**B**) Interactions between baobab polyphenols and the target proteins HMG-COA reductase and pancreatic lipase in 3D structures. (**C**) Interactions between baobab polyphenols and the target proteins HMG-COA reductase and pancreatic lipase in 3D structures.

**Figure 2 molecules-28-06112-f002:**
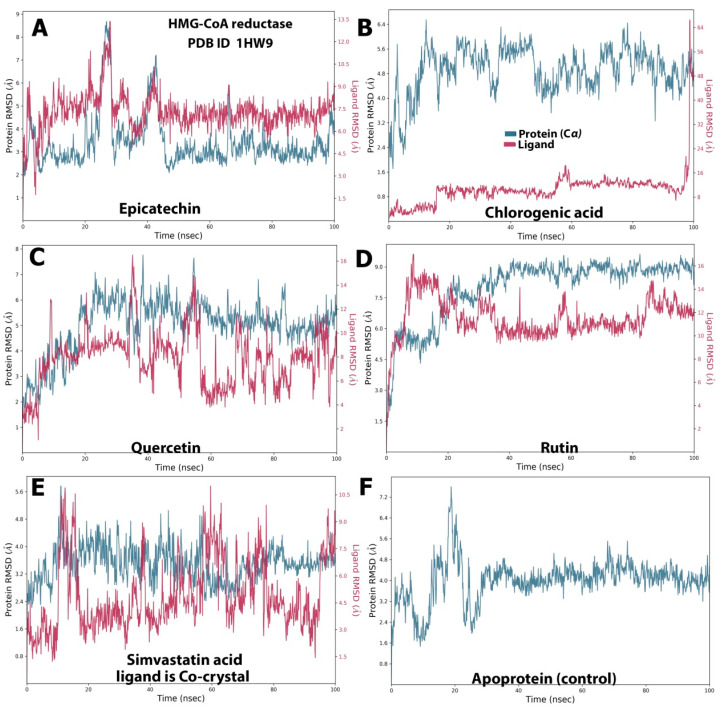
RMSD trajectory of the polyphenol–HMG-CoA reductase complex in the MD simulation analysis. The subfigures showed the RMSD values resulted from the interaction between (**A**) HMG-CoA reductase and epicatechin; (**B**) HMG-CoA reductase and chlorogenic acid. (**C**) HMG-CoA reductase and quercetin. (**D**) HMG-CoA reductase and rutin (**E**) showed interaction between HMG-CoA reductase and the standard ligand, simvastatin acid and (**F**) represented the RMSD value of the simulation of the protein alone without a ligand.

**Figure 3 molecules-28-06112-f003:**
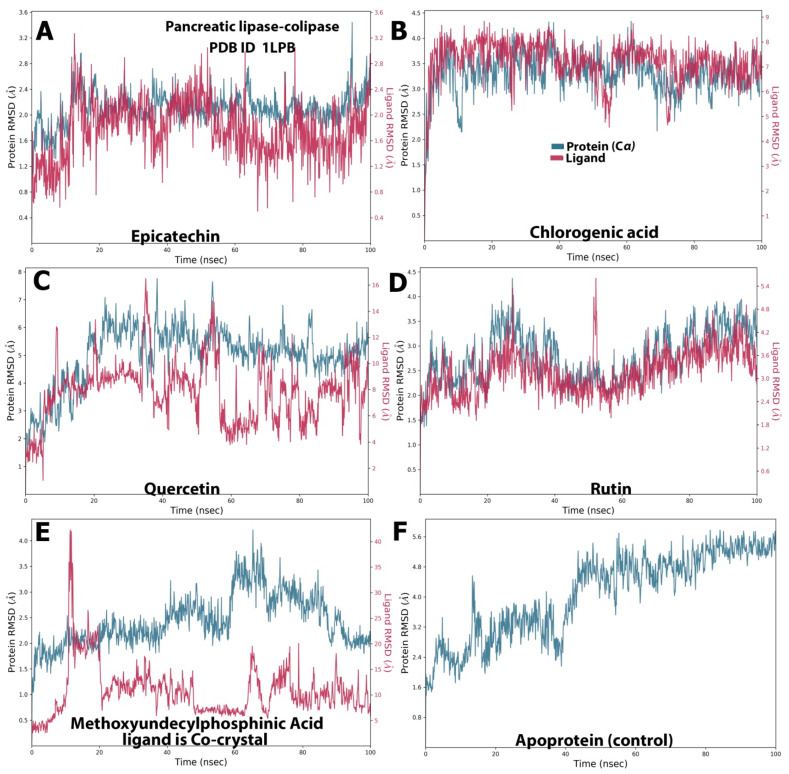
RMSD trajectory of the polyphenol–pancreatic lipase complex in the MD simulation analysis. The subfigures showed the RMSD values resulted from the interaction between (**A**) pancreatic lipase and Epicatechin; (**B**) pancreatic lipase and chlorogenic acid. (**C**) Pancreatic lipase and quercetin. (**D**) Pancreatic lipase and rutin. (**E**) Pancreatic lipase and the standard ligand, Methoxyundercylphosphinic Acid, and (**F**) Represented the RMSD value of simulation of the protein alone without a ligand.

**Figure 4 molecules-28-06112-f004:**
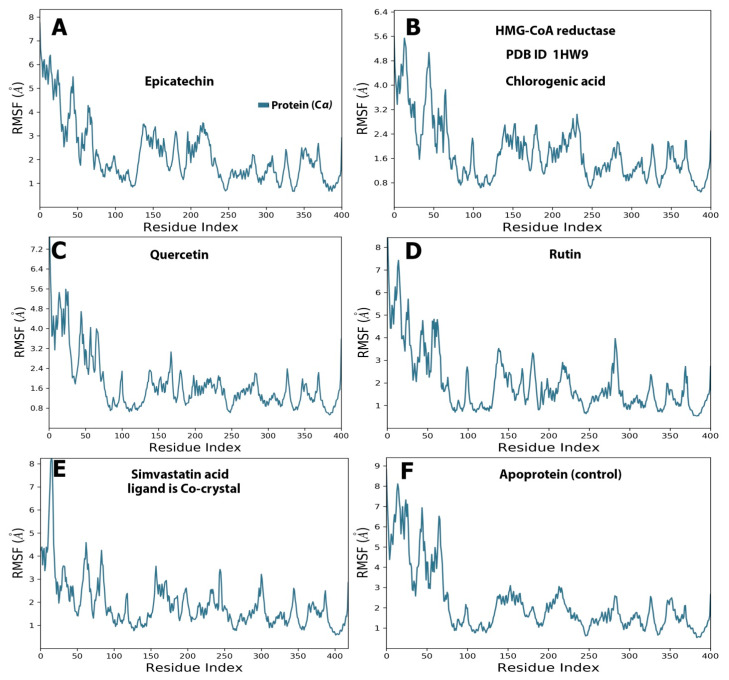
Selected compounds’ protein–ligand RMSF over 100 ns of the polyphenol-HMG-CoA reductase complex. The subfigures showed the RMSF values resulted from the interaction between (**A**) HMG-CoA reductase and epicatechin; (**B**) HMG-CoA reductase and chlorogenic acid. (**C**) HMG-CoA reductase and quercetin. (**D**) HMG-CoA reductase and rutin (**E**) showed interaction between HMG-CoA reductase and the standard ligand, simvastatin acid, and (**F**) represented the RMDF value of simulation of the protein alone without a ligand.

**Figure 5 molecules-28-06112-f005:**
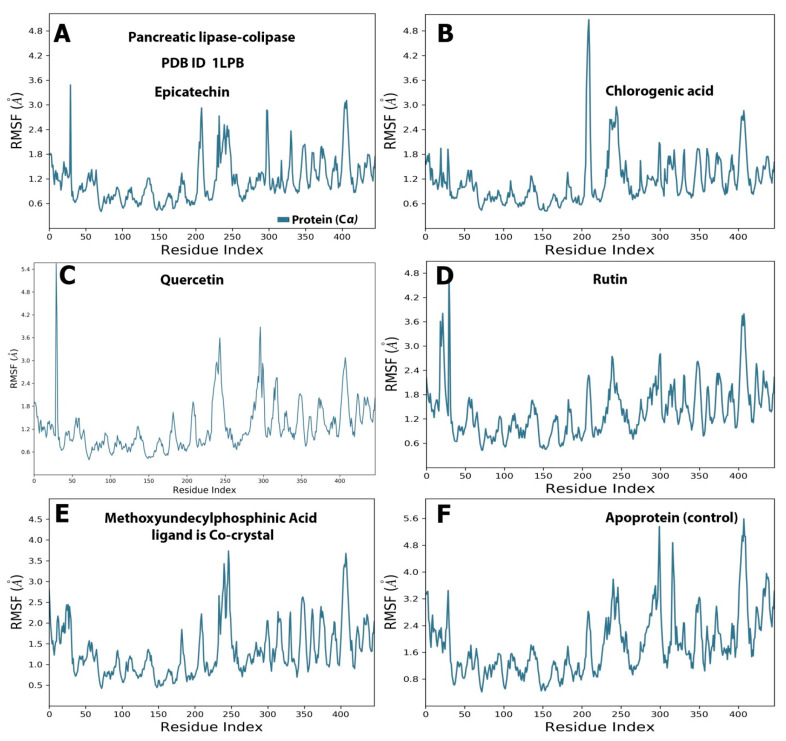
Selected compounds’ protein–ligand RMSF over 100 ns of the polyphenol–pancreatic lipase complex. The subfigures showed the RMSF values resulted from the interaction between (**A**) pancreatic lipase and Epicatechin; (**B**) pancreatic lipase and chlorogenic acid. (**C**) Pancreatic lipase and quercetin. (**D**) Pancreatic lipase and rutin. (**E**) Pancreatic lipase and the standard ligand, Methoxyundercylphosphinic Acid, and (**F**) Represented the RMSF value of the simulation of the protein alone without a ligand.

**Figure 6 molecules-28-06112-f006:**
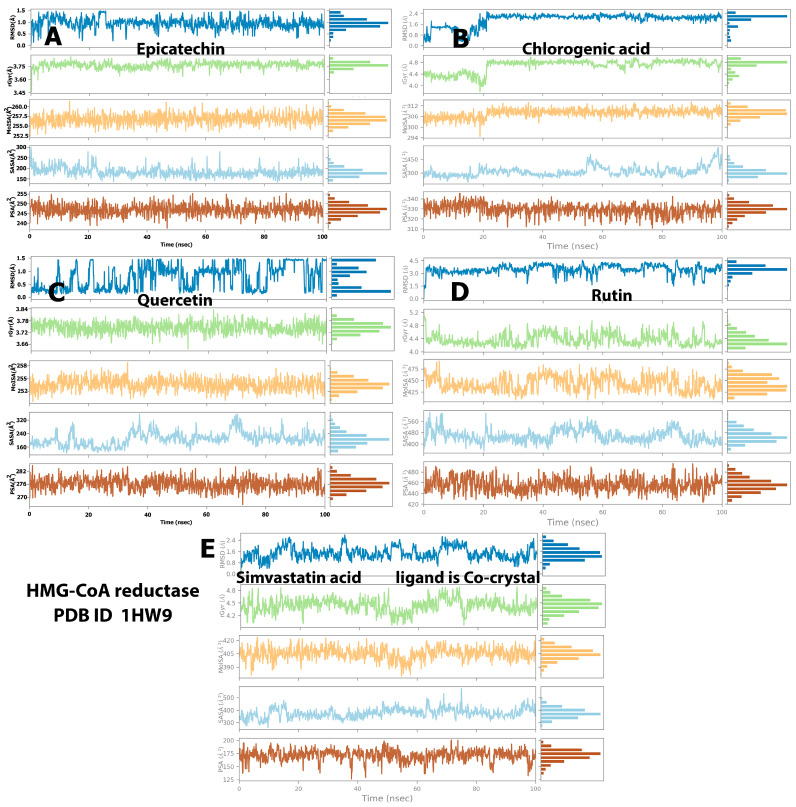
Postmolecular dynamic simulation analysis of HMG-CoA reductase complexes and ligand properties. (**A**) The RMSD, rGyr, MolSA, SASA and PSA values of epicatechin when simulated with HMG-CoA reductase during 100ns trajectory; (**B**) The RMSD, rGyr, MolSA, SASA and PSA values of chlorogenic acid when simulated with HMG-CoA reductase during 100ns trajectory. (**C**) The RMSD, rGyr, MolSA, SASA and PSA values of quercetin when simulated with HMG-CoA reductase during 100ns trajectory. (**D**) The RMSD, rGyr, MolSA, SASA and PSA values of rutin when simulated with HMG-CoA reductase during 100ns trajectory. (**E**) The RMSD, rGyr, MolSA, SASA and PSA values of the standard ligand, simvastatin acid when simulated with HMG-CoA reductase during 100 ns trajectory.

**Figure 7 molecules-28-06112-f007:**
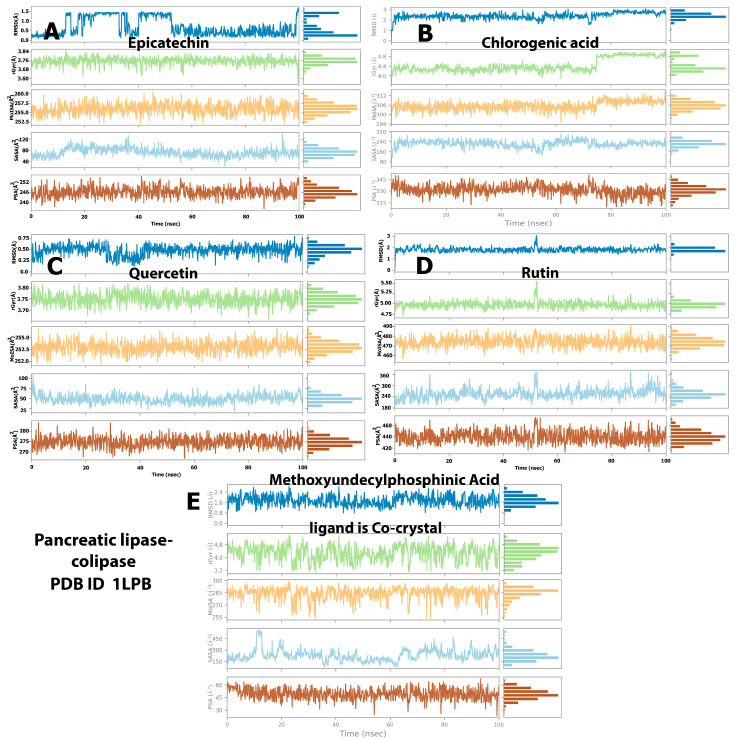
(**A**) Postmolecular dynamic simulation analysis of pancreatic lipase complexes and ligand properties. The RMSD, rGyr, MolSA, SASA and PSA values of epicatechin when simulated with pancreatic lipase during 100 ns trajectory; (**B**) The RMSD, rGyr, MolSA, SASA and PSA values of chlorogenic acid when simulated with pancreatic lipase during 100ns trajectory. (**C**) The RMSD, rGyr, MolSA, SASA and PSA values of quercetin when simulated with pancreatic lipase during 100ns trajectory. (**D**) The RMSD, rGyr, MolSA, SASA and PSA values of rutin when simulated with pancreatic lipase during 100 ns trajectory. (**E**) The RMSD, rGyr, MolSA, SASA and PSA values of the standard ligand, simvastatin acid when simulated with pancreatic lipase during 100 ns trajectory.

**Figure 8 molecules-28-06112-f008:**
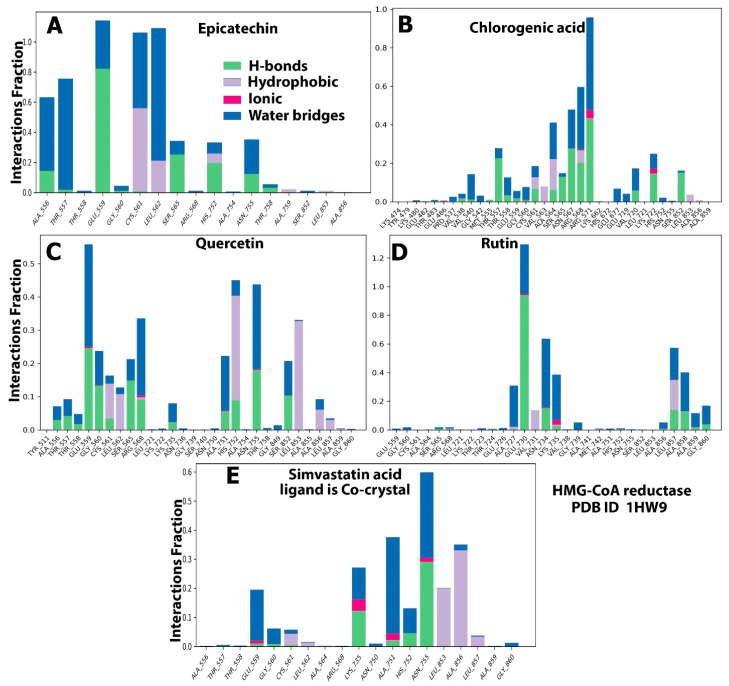
Histogram of the protein–ligand contacts of HMG-CoA reductase–baobab polyphenol complexes. (**A**) protein-ligand contact for epicatechin- HMG-CoA reductase complexes represented the bonding types and amino acid residues involved with the interaction fraction. (**B**) protein-ligand contact for chlorogenic acid-HMG-CoA reductase complexes represented the bonding types and amino acid residues involved with the interaction fraction (**C**) protein-ligand contact for quercetin- HMG-CoA reductase complexes represented the bonding types and amino acid residues involved with the interaction fraction. (**D**) protein-ligand contact for rutin- HMG-CoA reductase complexes represented the bonding types and amino acid residues involved with the interaction fraction. (**E**) protein-ligand contact for simvastatin acid- HMG-CoA reductase complexes represented the bonding types and amino acid residues involved with the interaction fraction.

**Figure 9 molecules-28-06112-f009:**
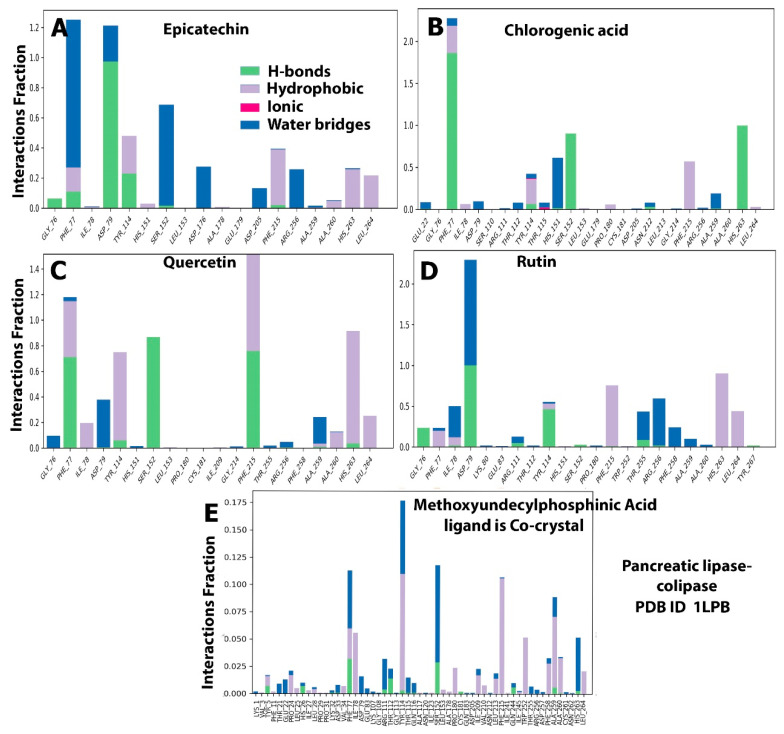
Histogram of the protein–ligand contacts of pancreatic lipase–baobab polyphenol complexes. (**A**) protein-ligand contact for epicatechin-pancreatic lipase complexes represented the bonding types and amino acid residues involved with the interaction fraction. (**B**) protein-ligand contact for chlorogenic acid- pancreatic lipase complexes represented the bonding types and amino acid residues involved with the interaction fraction (**C**) protein-ligand contact for quercetin-pancreatic lipase complexes represented the bonding types and amino acid residues involved with the interaction fraction. (**D**) protein-ligand contact for rutin- pancreatic lipase complexes represented the bonding types and amino acid residues involved with the interaction fraction. (**E**) protein-ligand contact for simvastatin acid- pancreatic lipase complexes represented the bonding types and amino acid residues involved with the interaction fraction.

**Figure 10 molecules-28-06112-f010:**
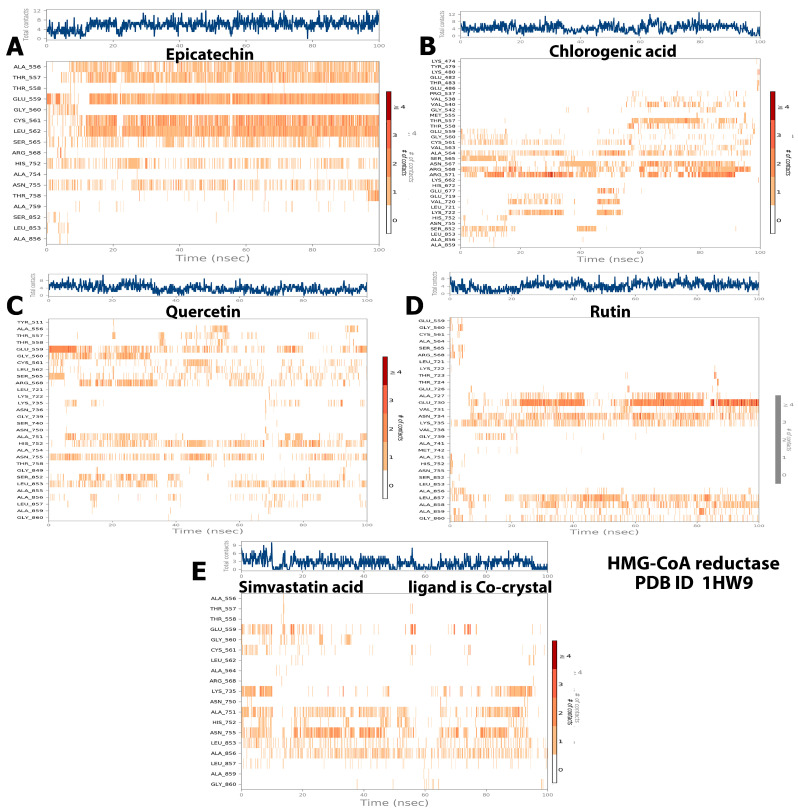
Tested compound–protein contact plot for baobab polyphenol–HMG-CoA reductase complexes during the simulation trajectory.

**Figure 11 molecules-28-06112-f011:**
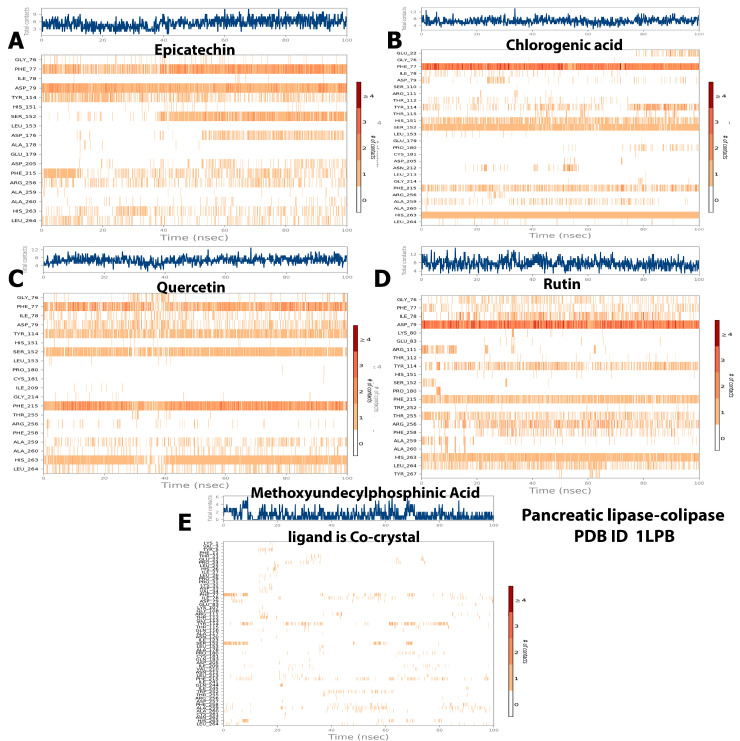
Compound–protein contact plot for baobab polyphenol–pancreatic lipase complexes during the simulation trajectory.

**Table 1 molecules-28-06112-t001:** Remains of interacting amino acids and binding bonds after docking.

Ligands	Affinity of the Binding (ΔG) kcal/mol against HMG-CoA Reductase	Interacting Amino Acid Residue(s) in HMG-CoA Reductase	Affinity of Binding (ΔG) kcal/mol against Pancreatic Lipase	Residue(s) of the Interacting Amino Acid in Pancreatic Lipase
Epicatechin	−5.5	GLY560, GLU559, GLY860, SER565	−9.4	PHE77, ASP79, HIS151
Chlorogenic acid	−5.7	GLU559, ALA564, SER852, CYS561	−8.6	PHE215, PHE77, ASP79, HIS263, HIS151 ARG256
Quercetin	−6.2	GLY560, HIS752, ALA564	−9.8	ASP79, ARG256, GLY76, HIS151
Rutin	−6.5	GLU559, ALA564, ALA751	−8.7	PHE215, ASP 9, ARG256, ALA259, HIS263
HMG-CoA cocrystal control	−5.2	GLU559, LYS735, ASN755	-	-
Pancreatic lipase cocrystal control MUP	-		−5.5	HIS263, PHE77

**Table 2 molecules-28-06112-t002:** (**A**) Binding energies (MM-GBSA) of the complexes of HMG-CoA reductase and baobab polyphenols. (**B**) Binding energies (MM-GBSA) of the complexes of pancreatic lipase and baobab polyphenols.

**(A)**								
**Compounds**	**MMGBSA dG** **Bind (Kcal/** **mol)**	**MMGBSA dG** **Bind Coulomb** **(kcal/mol)**	**MMGBSA dG** **Bind Covalent** **(kcal/mol)**	**MMGBSA dG** **Bind H-bond** **(kcal/mol)**	**MMGBSA dG** **Bind Lipo (kcal/mol)**	**MMGBSA dG** **Bind Packing** **(kcal/mol)**	**MMGBSA dG** **Bind Solv GB** **(kcal/mol)**	**MMGBSA dG** **Bind vdW** **(kcal/mol)**
**Epicatechin**	−41.241	−32.0133	0.917075	−2.49725	−10.038	−1.6385	28.47324	−24.445
**Chlorogenic acid**	−14.084	−11.7143	2.027484	−1.28119	−5.7528	24.95107	−22.3151	−14.0848
**Quercetin**	−25.014	−11.6902	1.259447	−1.39654	−4.0198	−2.91924	15.03979	−21.2879
**Rutin**	−42.749	−24.3292	2.892599	−2.17707	−8.19377	−2.98948	23.46127	−31.4134
**Simvastatin**	−31.62	26.47115	2.267489	−0.53712	−10.942		−20.7549	−28.132
**(B)**								
**Compounds**	**MMGBSA dG** **Bind (kcal/mol)**	**MMGBSA dG** **Bind Coulomb** **(kcal/mol)**	**MMGBSA dG** **Bind Covalent** **(kcal/mol)**	**MMGBSA dG** **Bind H-bond** **(kcal/mol)**	**MMGBSA dG** **Bind Lipo (kcal/mol)**	**MMGBSA dG** **Bind Packing** **(kcal/mol)**	**MMGBSA dG** **Bind Solv GB** **(kcal/mol)**	**MMGBSA dG** **Bind vdW** **(kcal/mol)**
**Epicatechin**	−34.773	−10.016	3.09688	−1.77298	−17.455	−1.51529	27.2685	−34.3795
**Chlorogenic acid**	−43.4258	−14.151	1.08367				18.8239	−27.5483
**Quercetin**	−57.9291	−18.634	1.8993	−1.599	−17.0719	−6.32072	28.23726	−42.5352
**Rutin**	−52.1559	−7.02153	0.030699	−2.05604	−18.4954	−3.51286	28.33565	−49.4365
**MUP**	−21.108	−0.5151	4.030		−16.112		17.85074	−21.09654

## Data Availability

The data presented in this study are available on request from the corresponding author.

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
