# Peer review of "Potential Effect of Baobab’s Polyphenols as Antihyperlipidemic Agents: In Silico Study"

_molecules, 2023, doi:10.3390/molecules28166112_

Round 1

Reviewer 1 Report

In the submitted manuscript Alameen et. al. investigated the mechanism of Adansonia Digitate L. polyphenols epicatechin, chlorogenic acid, quercetin, rutin, simvastatin acid, and methoxyundecylphosphinic acid on the activity of HMG-CoA reductase and pancreatic lipase, involved in lipid metabolism by applying molecular docking and molecular dynamics simulations.

The current manuscript requires major revisions before its publication. The following comments should be addressed:

Section Introduction

Line 47: The line break should be removed.

The new Figure of Adansonia Digitate L. would be very beneficial for the readers.

Lines 74-75: Known beneficial biological effects of polyphenolic compounds from Adansonia Digitate L. should be briefly described. The biological effects of polyphenolic compounds were comprehensively reviewed in recent article, which should be quoted and discussed.

Reference:

1.      Foods 2023, 12, 802.

Section Materials and Methods

2.1. Protein and ligand preparation and docking

Lined 98-101: How were the binding sites on HMG-CoA reductase and pancreatic lipase-colipase proteins identified?

2.2 MD simulations

How many Na+ and Cl- ions were added for the neutralization of the system?

The details of the applied thermalization and equilibration protocol should be provided and properly referenced.

MM/GBSA methodology to calculate binding free energies should be described.

Section Results and Discussion

To improve the overall readability of the manuscript, all figures should contain corresponding a), b), c), d), etc. labels when addressed in the main text.

The font size on the y- and the x-axis of all Figures 1-11 should be increased to be clearly visible.

There are some additional comments regarding the specific subsections:

3.1. Molecular docking study

The units of binding affinities should be added throughout the subsection.

Figure 1: I advise the authors to present both 2D figures in separate panel figure with the corresponding figure caption. Both 3D figures should be presented in a panel figure with the corresponding figure caption as well. The colors and types of depicted intermolecular interactions should be explained in figure captions. Moreover, the lengths of intermolecular interactions should be added.

3.3. Stability of protein-ligand complexes

Lines 177-179: The authors state:«On contrary, a major fluctuation was displayed by chlorogenic acid (2.00– 64.00 Å), while the protein showed narrow fluctuation between 1.6 and 6.4 Å.« According to the extremely large RMSD values presented in Figure 2 b), the chlorogenic acid left the binding site and the system likely exploded at the end of MD simulation. The authors should repeat the MD simulation for the chlorogenic acid-HMG-CoA system at different random seeds. 

As can be observed from Figure 2, very broad oscillations in RMSD and RMSF were obtained, which indicate the repositioning of ligands at the binding site and conformational changes within the protein. This instability is also reflected in the calculated binding free energies of the complexes (Table 2a and 2b).

To address this issue, the authors should perform the MD simulations in multiple parallels instead of running only one long MD simulation. Moreover, I suggest the authors apply more rigorous binding free energy calculations based on the Linear interaction energy method (LIE) and Linear Response Approximation (LRA), which provide better approximations of the experimental binding free energies. Quote and discuss.

Reference:

1.       Antioxidants 2023, 12, 63.

3.4. Protein and ligand properties from MD simulation analysis

Line 255: The discussed results are presented in Figure 6, not Figure 4.

Lines 263-265: The authors state:« However, the last ligand, Quercetin, showed different behavior and massive fluctuation ranging from 25 to 60 Å with equilibrium at 50 Å (Figure7).«

According to the extremely large RMSD values presented in Figure 7, quercetin left the binding site. The authors should repeat the MD simulation for the quercetin-pancreatic lipase system at different random seeds. 

Figure 6: Bold font should be corrected into the normal font.

Figure 7: The protein name »HMG-CoA reductase« should be corrected to »pancreatic lipase« in the figure caption.

 3.5. Protein–ligand interaction and bonds formation

The interaction fractions of detected intermolecular interactions with specific amino acids should be provided at the corresponding positions in the main text.

Which geometrical criteria were used to define intermolecular interactions?

Figure 11: The protein name »HMG-CoA reductase« should be corrected to »pancreatic lipase« in the figure caption.

Section Conclusion

Line 395: Very recently new inverse molecular docking protocol was proposed to identify potential protein targets of polyphenols from Adansonia Digitate L., which could also provide further insights into their molecular mechanisms in different lipid metabolic pathways. Quote and discuss.

Reference:

- Foods 2022, 11, 1253.

OK.

Author Response

Dear Doctor,

We correct the majority of your comments, but some are tough to change; we hope you understand.

Reviewer 2 Report

The authors reported a theoretical study about the potential effect of four polyphenolic compounds of the Baobab’s tree as antihyperlipidemic agents.

After a detailed revision of the manuscript, the following observations were made:

 1.    It is not clear why these polyphenols were specifically selected for the study. I recommend include more references about this.

 2.    Molecular docking is not the same as molecular dynamics, it is important to correct this term in line 103, methodology section, and therefore include the term in the abstract.

 3.    In lines 30-31, abstract section, “Moreover, the analysis of RMSF simulation results and RMSD 30 analysis showed a high degree of consistency”, is not clear. It should be more explained, specifically: a high degree of consistency.

 4.    Check the PubChem ID: 72 (in line 92, methodology section), it does not match with Methoxyundecylphosphinic acid. Please review this information.

 5.    It is mentioned that images of HMG-CoA reductase and pancreatic lipase-colipase were created by ChemDraw (lines 92-93), however, it was not mentioned what they were used for, and also were not discussed in the results. As far as I know, ChemDraw software does not draw 3D figures. Please check this information.

 6.    The PDB ID: 2OBI, it does not correspond to studied enzymes in the present work. Line 97. Please check this information.

 7.    The molecular dynamics methodology needs to be more detailed, for example: how the protein and ligands were parametrized. It was not mentioned the unit of size orthorhombic box (what do you mean by 10?).

 8.    The figures are supersaturated and are not correctly described. These figures must be redesigned and a detailed figure caption must be included. The quality of the figures should be improved.

 9.    The interactions between HMG-CoA reductase and the polyphenols: chlorogenic acid and quercetin have already been studied by molecular docking and dynamics, it is important to highlight the difference or main contribution in this paper about the interactions between this enzyme and these specific polyphenols compared to other reported works.

Extensive editing of English language required.

There are words that should be written in italics such as: in silico, adansonia digitate L., in vivo, in vitro and et al.

Be sure to check for spelling errors, typos and incorrect punctuation, for example line 23, 5th word, line 25, 10th word, spaces in the text, capital letter in word “All” in line 138, check redaction of sentence in line 112, etc.

Check redaction of methodology: Protein and ligand preparation and docking.

In line 184, change MRSD to RMSD.

Homogenize font size such as in line 267 caption figure. Please check all the text.

Author Response

Dear Doctor,

We correct the majority of your comments, but some are tough to change; we hope you understand. for the English language we already edited by expert doctor.

Round 2

Reviewer 1 Report

The authors successfully addressed the majority of issues raised by this Reviewer. Consequently, the manuscript has been significantly improved and can be in its current version recommended for publication in Molecules.

OK.

Author Response

Sincere gratitude for your decision

Reviewer 2 Report

The authors reported the second version of the manuscript titled “The potential effect of Baobab’s polyphenol as antihyperlipidemic agent: in silico study”.

After a detailed revision of the resubmitted version of manuscript, the following observations were made:

According to the response to the reviewers, the authors have only addressed 3 points of the suggested changes in the first revision, while some of the suggested changes were related to: include more references about the analyzed polyphenols, include a more detailed methodology, more explanation about the software employed to generate images, and to highlight the difference or main contribution in this paper about the interactions between this enzyme and these specific polyphenols compared to other reported works.

In my opinion, these changes are not difficult to address; however, they are merely suggestions made by this reviewer to enhance the presentation of their results. It is likely that all they need is a little more time to respond to the suggestions. 

I recommend a revision of the English by a grammar expert, in my opinion a moderate editing of English language is required.

Author Response

all new change highlighted by yellow.

Round 3

Reviewer 2 Report

Dear authors.

After carefully reviewing the latest version of your article, I appreciate that you have addressed my suggestions for the improvement of your paper.

I have already made my decision and communicated it to the editor.

Thank you